# From Body Monitoring to Biomolecular Sensing: Current Progress and Future Perspectives of Triboelectric Nanogenerators in Point-of-Care Diagnostics

**DOI:** 10.3390/s24020511

**Published:** 2024-01-14

**Authors:** Zequan Zhao, Yajun Mi, Hafeez Ur Rehman, Enqi Sun, Xia Cao, Ning Wang

**Affiliations:** 1Center for Green Innovation, School of Mathematics and Physics, University of Science and Technology Beijing, Beijing 100083, China; m202110789@xs.ustb.edu.cn (Z.Z.); d202110423@xs.ustb.edu.cn (Y.M.); hafeezurrehmandgk59@gmail.con (H.U.R.); sunaadk@163.com (E.S.); 2Beijing Institute of Nanoenergy and Nanosystems, Chinese Academy of Sciences, Beijing 100083, China; 3School of Chemistry and Biological Engineering, University of Science and Technology Beijing, Beijing 100083, China

**Keywords:** point-of-care diagnostics, triboelectric nanogenerator, self-powered sensing, wearable electronics

## Abstract

In the constantly evolving field of medical diagnostics, triboelectric nanogenerators (TENGs) stand out as a groundbreaking innovation for simultaneously harnessing mechanical energy from micromovements and sensing stimuli from both the human body and the ambient environment. This advancement diminishes the dependence of biosensors on external power sources and paves the way for the application of TENGs in self-powered medical devices, especially in the realm of point-of-care diagnostics. In this review, we delve into the functionality of TENGs in point-of-care diagnostics. First, from the basic principle of how TENGs effectively transform subtle physical movements into electrical energy, thereby promoting the development of self-powered biosensors and medical devices that are particularly advantageous for real-time biological monitoring. Then, the adaptable design of TENGs that facilitate customization to meet individual patient needs is introduced, with a focus on their biocompatibility and safety in medical applications. Our in-depth analysis also covers TENG-based biosensor designs moving toward exceptional sensitivity and specificity in biomarker detection, for accurate and efficient diagnoses. Challenges and future prospects such as the integration of TENGs into wearable and implantable devices are also discussed. We aim for this review to illuminate the burgeoning field of TENG-based intelligent devices for continuous, real-time health monitoring; and to inspire further innovation in this captivating area of research that is in line with patient-centered healthcare.

## 1. Introduction

The development of the triboelectric nanogenerator (TENG) marks a significant milestone in the field of energy harvesting and sensory applications due to the coupled effects of triboelectricity and electrostatic induction [1,2,3,4]. This innovative device is capable of converting mechanical energy from various sources, such as human motion, wind, or water waves, into useful electrical energy [5,6,7]. This conversion principle, which is grounded in the triboelectric effect, involves a wide selection of triboelectric materials with different and, in some cases, even the same, electron affinity that lead to electron transfer and generate an alternating electrical current if combined with the electrostatic induction effect [8,9,10]. Consequently, the broad applicability and adaptability of TENGs make them a sustainable and efficient solution in diverse sectors, ranging from environmental monitoring to biomedical applications. This is especially true in settings where conventional power sources are impractical or unavailable [11,12,13,14]. This versatility, coupled with their environmental friendliness and cost effectiveness, positions TENGs at the forefront of next-generation technology for energy harvesting and self-powered sensing systems [15,16,17].

Point-of-Care Diagnostics (PoCD) have revolutionized medical testing by bringing diagnostic tools directly to the patient’s location [18,19,20,21]. This approach significantly reduces the time between testing and obtaining results, which is crucial in many clinical scenarios. The main benefits of PoCD include portability, ease of use, and rapid turnaround time, allowing for immediate medical decision making [22,23,24,25]. These diagnostics cover a wide spectrum, from glucose monitoring in diabetes management to infectious disease testing. Their deployment in various settings, including remote areas and in-field operations, underscores their versatility and their critical role in enhancing global health access. By simplifying and expediting the diagnostic process, PoCD serves as a vital tool in managing and controlling diseases, particularly in resource-limited environments, where traditional laboratory facilities are scarce.

Despite the significant advancements in, and benefits of PoCD, several challenges hinder their widespread adoption and effectiveness. The primary issue is their reliance on stable power sources, which is a significant barrier in remote or underdeveloped areas. Furthermore, the accuracy and sensitivity of these devices may be compromised due to their miniaturized and portable nature; potentially leading to less reliable results compared to standard laboratory equipment. Another critical aspect is the need for user-friendly interfaces, as these devices often operate outside of traditional clinical settings and are used by individuals without specialized training. Ensuring robustness and durability is also paramount, as PoCD devices must withstand varied environmental conditions without degradation of performance. Additionally, the integration of data management systems that securely and efficiently handle patient information is crucial, especially with the increasing emphasis on telemedicine and remote health monitoring. Addressing these challenges is essential for realizing the full potential of PoCD in transforming healthcare delivery, particularly in areas where access to conventional medical facilities is limited.

The integration of TENG into PoCD systems presents a novel solution to many of these challenges. Firstly, TENGs offer a sustainable and reliable power source that is crucial for PoCD’s operation in remote or power-limited environments. The ability of TENG to harness energy from everyday movements and environmental factors negates the dependence of PoCD systems on external power supplies [26,27,28]. This aspect not only enhances their portability, but also significantly increases the accessibility of PoCD devices in various settings. Moreover, TENG technology can potentially improve the sensitivity and accuracy of diagnostic tools through advanced sensor integration [29,30,31,32]. This enhancement is critical in achieving laboratory-level precision in portable formats. Additionally, TENGs can contribute to the durability and robustness of PoCD devices due to their inherent mechanical strength and flexibility, considering the wide selection of materials and various working modes. This feature is particularly beneficial in rugged or harsh environments, as it ensures consistent device performance. The adaptability of TENGs also allows for customization in PoCD design, accommodating diverse medical applications and user requirements [33,34,35]. This flexibility may pave the way for more user-friendly interfaces that are tailored to the specific needs and skills of the end-user. Lastly, TENGs can play a pivotal role in advancing data management systems within PoCD, facilitating efficient and secure transmission of medical data; this is integral to telemedicine and remote healthcare services.

The advantages of TENGs in PoCD are multifaceted and go beyond their self-powering capability [36,37,38]. For instance, the flexibility of TENGs allows for the creation of conformable and wearable diagnostic devices that can comfortably adhere to various body parts, providing continuous health monitoring. This adaptability is especially beneficial in developing patient-specific diagnostic tools. Moreover, the customizability of TENGs enables the design of devices tailored to specific medical conditions, enhancing diagnostic precision and efficacy. In terms of biocompatibility, TENG materials can be engineered to be non-toxic and compatible with the human body, ensuring they are safe and suitable for long-term use [39,40,41,42]. This feature is critical in applications involving them being in direct contact with skin or used internally. The low cost of TENG components and their manufacturing process significantly reduces the overall expense of diagnostic devices, making PoCD affordable and accessible to those in low-resource settings. Furthermore, TENGs exhibit high durability and reliability, which are essential for consistent device performance over time [43,44]. Their ability to operate effectively in diverse environmental conditions further broadens the scope of their application in global health contexts. The integration of TENGs in PoCD not only addresses current limitations, but also opens new possibilities for innovative, efficient, and patient-centric diagnostic solutions.

In this review, we comprehensively explore the integration of TENGs in PoCD systems, focusing on their application in various medical fields (Figure 1). We delve into the fast advancements that TENG technology has brought to neurological system diagnosis, such as in the monitoring and management of Parkinson’s disease. The review highlights the development of innovative, adaptable, and biocompatible TENG-based devices, demonstrating their effectiveness in capturing subtle, yet crucial, diagnostic information. Furthermore, we explore the role of TENGs in musculoskeletal system diagnostics, including sophisticated tools for monitoring finger motion and fall detection that are crucial for patient care in elder care and rehabilitation. We also examine the use of TENGs in bone morphology diagnosis, emphasizing their non-invasive nature and potential for remote monitoring applications. The review extends to the realm of biochemical sensing, showcasing how TENGs are revolutionizing glucose and protein sensing, thereby aligning with trends toward more personalized and patient-centered healthcare approaches. This comprehensive analysis underscores the versatility and potential of TENGs in enhancing PoCD across various medical specialties and promising a future where healthcare is more accessible, personalized, and efficient.

## 2. Working Principle of TENG

TENGs primarily harness the triboelectric effect and electrostatic induction to convert biomechanical energy into electrical energy, which can then power medical equipment. The basic principle is that when two materials with differing electronegativity come into contact, electron transfer occurs between them [45]. Upon separation, the electrostatic induction effect causes electrons to flow through an external load. Repeating this process enables TENGs to generate alternating current. Based on this, TENGs are categorized into four working modes (Figure 2) [10]: vertical contact-separation mode [46,47], lateral sliding mode [48,49], single electrode mode [50,51,52], and freestanding triboelectric layer mode [53,54].

Vertical contact-separation (CS) mode: In this mode, when two objects with different electronegativities are in vertical contact, electron exchange occurs at the contact surface. As they separate, the equal but opposite charges between them induce a potential difference across attached electrodes, generating an electric current when they are connected through wires. This potential difference diminishes over time. Upon re-contact, the process reverses, creating an opposite potential difference and current. This cycle of contact and separation produces alternating current.

Lateral sliding (LS) mode: This mode operates similarly to the vertical contact-separation mode, but with a key difference: the relative displacement shifts from vertical to horizontal. The alternating current is generated through repeated horizontal displacement.

Single electrode (SE) mode: The simplest in structure, this mode utilizes the ground as an electrode. A potential difference is generated between a metal electrode and the ground through electrostatic induction, thereby inducing current flow.

Freestanding triboelectric layer (FT) mode: In the FT mode, a charged object is placed between two electrodes that are attached to dielectric layers. Movement of the charged object between these electrodes alters the potential difference, resulting in current generation.

## 3. TENG for Cardiovascular and Respiratory System Real-Time Diagnosis

### 3.1. Cardiovascular Diagnosis

In recent years, cardiovascular disease (CVD) has emerged as a leading cause of morbidity and mortality worldwide. Early and accurate diagnosis is critical for effective management and treatment of CVD. Traditional diagnostic methods, while effective, often require complex equipment and can be invasive. The advent of TENG technology presents a transformative opportunity in this field and enables the development of self-powered, wearable diagnostic devices. This section delves into the advancements in TENG technology for cardiovascular diagnosis with a focus on cuff-free blood pressure monitoring and pulse diagnosis (Table 1). The integration of TENGs into wearable technology not only promises to enhance diagnostic accuracy but also to revolutionize the current approach to patient-centered care in cardiology.

#### 3.1.1. Blood Pressure and Pulse Diagnosis

In the field of cardiovascular diagnostics, particularly at the point-of-care level, the integration of TENG technology has marked a paradigm shift, especially in blood pressure monitoring techniques. A significant leap in this domain has been the introduction of a cuff-free, self-powered continuous blood pressure monitoring system by Ran et al. [55] (Figure 3a). This system is a testament to the innovative application of TENG technology in medical devices. The core of this system is a double sandwich-structured triboelectric sensor, where the thoughtfully engineered composition of the electrode materials not only elevates the system’s sensitivity but also reinforces its structural integrity, ensuring long-term durability. The sensor’s advanced design showcases remarkable adaptability and precision in detecting blood vessel signals. Its high sensitivity, quantified at 0.89 V/kPa, is a clear indicator of its ability to detect minute changes in blood pressure. Furthermore, its rapid response time of just 32 ms underscores the system’s efficiency in real-time monitoring; this is a critical factor in managing cardiovascular conditions. In the context of PoCD, this TENG-based blood pressure monitoring system stands out for its user-friendliness and non-invasive nature. Its ability to provide continuous, real-time data is vital for the effective management of cardiovascular diseases, especially in settings where immediate medical attention is paramount [61]. Furthermore, the self-powered characteristic of this technology circumvents the need for external power sources, making it exceptionally suitable for use in remote or resource-limited environments.

Pulse diagnosis stands as a critical component in the realm of cardiovascular health assessment, and the integration of TENG technology has marked a significant evolution in this area. Among the notable advancements, the work of Xu et al. in developing a self-powered sensitive ultra-pulse sensor (SUPS) represents a milestone in enhancing pulse diagnosis capabilities (Figure 3b) [56]. This innovative SUPS incorporates a nanowire array that is complemented by a melamine sponge layer, which empowers the SUPS with exceptional sensitivity and responsiveness. The device exhibits a remarkable sensitivity of 10.29 nA/kPa, making it highly capable of detecting even the most subtle changes in pulse rhythm. This level of sensitivity is crucial for identifying and monitoring various cardiovascular conditions, such as atrial fibrillation and supraventricular tachycardia, which require precise and continuous pulse monitoring. Additionally, the SUPS’s rapid response time of just 30 ms underscores its efficiency in real-time data acquisition, an essential feature for timely medical interventions. The SUPS stands as a testament to the potential of TENG technology to revolutionize medical diagnostics. By enabling non-invasive, long-term monitoring of cardiovascular health, this device not only enhances patient comfort but also provides medical professionals with accurate and consistent data for better patient management. Its ability to monitor cardiovascular rhythms continuously makes it a valuable tool in both clinical settings and for home health monitoring, offering a new level of accessibility and convenience in cardiac care.

Wang et al. have developed a TENG system that leverages the properties of biocompatible polyvinyl alcohol (PVA) blends, as detailed in (Figure 3c) [57]. Uniquely adaptable, the PVA blend film can be modified with various fillers, such as gelatin, HCl/NaOH, or KCl/NaCl. This helps to optimize its dielectric properties and triboelectric performance. This advancement in TENG technology is pivotal for PoCD, as it efficiently converts mechanical energy into electrical power to facilitate self-powered health monitoring and therapeutic applications. The optimized PVA-gelatin blend, in particular, showcases stable electricity outputs and a low detection threshold for mechanical deformations such as the human pulse; this marks a significant step forward in developing cost-effective, biocompatible TENG devices for PoCD.

Further emphasizing the role of TENG in PoCD, Wang et al. introduced a groundbreaking self-powered, real-time sensing TENG vascular graft device, as shown in (Figure 3d) [58]. In this design, the electrospun poly(3-hydroxybutyrate)(PHB) membrane serves as the substrate, while the ePTFE membrane provides protective packaging. Equipped with copper electrodes for efficient charge collection and transfer, this device is adept at detecting various hemodynamic conditions and fostering the regeneration of endothelial and vascular tissues. Notably, its exceptional pressure sensitivity, particularly in simulated pulsatile blood flow experiments, underlines its potential for revolutionizing PoCD through real-time monitoring and early detection of vascular disorders.

#### 3.1.2. Cardiac Diagnosis

The development of TENG-based cardiac real-time diagnosis represents a significant advancement in the monitoring of heart activity. Utilizing the principles of TENGs, this approach enables the sensing and analysis of various cardiovascular functions. The strength of this technology lies in its modularity, which allows for the creation of devices that are not only compact and flexible but also biocompatible. These attributes greatly enhance the functionality and integration of devices within the cardiac monitoring and treatment landscape.

A prime example of this modularity in action is the self-powered endocardial pressure sensor (SEPS), which has been innovatively designed by Liu et al. [59]. The SEPS is a testament to the effectiveness of modular design, and comprises four distinct layers that collectively offer robust protection against blood and moisture (Figure 3e). This design facilitates the real-time monitoring of endocardial pressure, making it a critical tool in cardiac health management and allowing for seamless integration with surgical catheters. This feature is particularly advantageous for minimally invasive implantation, which is a key consideration in modern medical procedures. In clinical settings, the SEPS proves invaluable for patients with heart failure, offering crucial data that can guide treatment decisions.

In the realm of PoCD, the innovation by Zhao et al. stands out for the creation of a no-spacer TENG (NSTENG). This is specifically engineered for cardiovascular monitoring, as depicted in Figure 3f [60]. This system’s eco-friendly and in situ gap generation approach not only safeguards biological safety but also mitigates the risk of air contamination, which is a significant consideration in point-of-care settings. The NSTENG excels in monitoring both standard and atypical heart activities, providing a more expansive scope of cardiac analysis compared to traditional electrocardiograms. Its dual capability as both a wearable and implantable device makes it an invaluable asset in PoCD, as it is capable of detecting subtle cardiac nuances potentially missed by conventional ECGs. This advancement underscores the crucial role of NSTENG in the evolution of biosafe sensors and its transformative impact on enhancing cardiac monitoring within PoCD.

### 3.2. Respiratory System Diagnosis

Respiratory diseases, including conditions such as asthma, chronic obstructive pulmonary disease (COPD), and respiratory infections, represent a major global health burden. Timely and accurate diagnosis of these conditions is crucial for effective treatment and management. Traditional diagnostic methods, though reliable, can be invasive and are not always feasible in remote or resource-limited settings. The advent of TENG technology in respiratory system diagnosis offers a promising alternative. This section explores the role of TENG-based devices in revolutionizing respiratory diagnostics (Table 2). These devices leverage the principles of energy harvesting from body movements and enable non-invasive, real-time monitoring of respiratory parameters. In this section, the latest advancements in TENG technology for hypoventilation and asthma diagnostics will be reviewed; and their potential to provide portable, user-friendly, and accurate respiratory health monitoring solutions will be highlighted. This technological evolution not only enhances patient care but also extends the reach of respiratory diagnostics to remote and underserved areas, paving the way for more inclusive and accessible healthcare.

#### 3.2.1. Respiratory System Diseases Diagnosis

Hypoventilation diagnostics are crucial for detecting respiratory issues, particularly in conditions such as hypoventilation syndrome and asthma. The integration of TENG technology has led to significant advancements in this area, enhancing the capability for accurate and real-time respiratory monitoring.

The advancements in PoCD for respiratory conditions have been significantly enhanced by the development of a wearable medical device by Peng et al. [62], which was tailored specifically for respiratory diseases, including hypoventilation syndrome and asthma. In this work, a modular electronic skin (e-skin) was developed for real-time respiratory monitoring and diagnosing obstructive sleep apnea-hypopnea syndrome (OSAHS) (Figure 3g). This e-skin uses TENG technology and consists of multilayer polyacrylonitrile and polyamide 66 nanofibers, along with gold electrodes. The e-skin is capable of providing energy autonomy and accurate real-time respiration monitoring. Its high-pressure sensitivity, good air permeability, and excellent working stability make it a suitable foundation for a self-powered diagnostic system. This system can detect and evaluate the severity of OSAHS in real time, thereby potentially improving sleep quality. In the context of PoCD, the application of TENG technology in hypoventilation diagnostics marks a significant advancement because they can provide healthcare professionals with immediate and actionable data, enabling prompt intervention in acute respiratory conditions. Their ease of use and non-invasive nature make them ideal for various settings, including home-based care, thereby enhancing patient accessibility to critical health monitoring.

Asthma diagnostics play a crucial role in respiratory healthcare, and the integration of TENG technology, has introduced innovative approaches for asthma management. These advancements are particularly significant for detecting and monitoring asthma symptoms and triggers.

Yu et al. have made further advancements in TENG-based respiratory monitoring by developing a wearable point-of-care (POC) medical device specifically for respiratory diseases such as asthma [63]. This device comprises a self-switched TENG (SS-TENG) and a transdermal drug delivery system (Figure 3h). The SS-TENG generates electrical energy from the wearer’s respiratory motions. It operates in two modes: prevention and emergency. The modular architecture of this device allows for flexibility in response to respiratory motion and can issue both visual and wireless emergency alarms, making it a valuable tool for continuous monitoring and prevention of respiratory diseases. In PoCD, TENG technology enhances asthma management by enabling real-time monitoring of respiratory health. This capability is essential for adjusting treatment plans and preventing severe asthma attacks. The portability and ease of use of TENG-based devices make them suitable for home monitoring, empowering patients to manage their condition more effectively.

The utilization of TENG-based devices for hypoventilation diagnostics can transform the management of respiratory diseases. By enabling real-time monitoring of respiratory patterns, these devices aid in the early detection of potential respiratory failures or exacerbations in conditions such as asthma and OSAHS [67,68,69]. This proactive approach may lead to better disease management and improved patient outcomes, and potentially reduce the burden on healthcare systems.

#### 3.2.2. Respiratory Product Detection

In the realm of respiratory product detection, several innovative studies have been conducted, focusing on the sensing of various gases such as carbon dioxide, ammonia, and ethanol, which are crucial indicators of respiratory health and environmental quality.

Zhao et al. have developed a groundbreaking self-powered CO_2_ gas sensor leveraging the triboelectric nanogenerator (TENG), revolutionizing CO_2_ detection in the context of the Internet of Things (IoT) [64] (Figure 3i). This innovative sensor operates at room temperature and is characterized by its high sensitivity and independence from external power sources. Its working principle is based on the distinctive behavior of gas discharge induced by TENG. Specifically, when CO_2_ is mixed with N_2_, the produced negative CO_2_ ions interfere with plasma formation, thereby increasing the threshold voltage for gas discharge and altering its characteristics. This phenomenon underpins the sensor’s diverse detection modes, including a threshold concentration detection mode in which the discharge ceases at a specific CO_2_ concentration. The sensor’s capability is further exemplified by its ability to adjust detectable threshold concentrations between 1000 and 200,000 ppm and to utilize step and continuous detection modes for CO_2_ concentrations below the threshold. Additionally, the sensor showcases unique discharge electrode distances (dmax) for different gases, with CO_2_, O_2_, air, and N_2_ measuring 0.11, 0.28, 0.45, and 0.55 mm, respectively. These features enable the sensor to detect CO_2_ concentrations with exceptional precision, as demonstrated by the linear increase in discharge current with rising CO_2_ levels until the threshold concentration is reached, at which point the discharge current drops to zero.

In this comprehensive study, Zheng et al. have innovatively designed a self-powered CO_2_ gas detection method, leveraging a rotating TENG with a needle-like ionization structure (Figure 3j) [65]. This unique approach facilitates the ionization of CO_2_ gas, significantly enhancing the output performance of the TENG system under varying conditions of air pressure, concentration, and humidity. The research demonstrates that the TENG’s voltage output varies linearly with changes in CO_2_ gas pressure, from 10 kPa to 60 kPa; and concentration, from 10% to 90%. Additionally, humidity tests reveal a complex relationship with voltage change, which turns from a negative to a positive correlation around the 55% RH mark. The TENG’s high-performance output, which is driven by the ionization of gas at the needle tips, is instrumental in distinguishing CO_2_ gas characteristics quickly and accurately. This method stands out for its simplicity, cost effectiveness, and self-powered operation, making it highly suitable for industrial and IoT applications. The concept was inspired by ionization phenomena in lightning strikes, leading to the development of a high electric field between two needle tips connected to the TENG’s friction layer. Unlike conventional CO_2_ sensors that rely on oxide semiconductors, optical, or electrochemical methods—which all require external power and suffer from high power consumption and long detection times—this innovative method operates efficiently at room temperature with minimal energy input. The TENG’s design ensures a stable structure and regular output signal, providing a convenient and effective means for dynamic CO_2_ monitoring.

In parallel, the field of ammonia sensing has witnessed a groundbreaking development through the work of Veeralingam et al. They introduced an integrated gas sensing system that synergistically combines a triboelectric nanogenerator (TENG) with a highly sensitive gas sensor (Figure 3k) [66]. This innovative system utilizes Titanium (Ti) functionalized Molybdenum Disulfide (MoS_2_) meticulously interspersed within a matrix of polypropylene and Nylon cloth, creating a composite material with unique sensing properties. The dual functionality of this system is noteworthy; it acts not only as a highly sensitive respiration sensor but also as a self-powered ammonia gas sensor. This dual capability is particularly significant, as it opens new avenues for health monitoring and diagnostic applications where lack of power availability is a constraint. The system demonstrated exceptional performance in detecting ammonia at room temperature, which is a testament to the effectiveness of Ti@MoS2 nanoparticles. This feature is crucial for health diagnostic applications, especially in settings where traditional power sources are unavailable or unreliable. The success of this system in ammonia detection underlines the potential of integrating TENG technology with gas sensing, paving the way for the development of more versatile, efficient, and self-sustaining health monitoring devices.

## 4. TENG for Neuromuscular System Real-Time Diagnosis

This section shifts focus to the application of TENG technology in diagnosing neurological and musculoskeletal systems. These are areas in which precision and sensitivity in diagnostics are particularly crucial (Table 3). Neurological disorders, such as Parkinson’s disease, present unique challenges due to their complex symptoms and progression patterns. Similarly, in musculoskeletal diagnostics, TENGs are instrumental in developing advanced tools for monitoring activities such as finger motion and detecting falls, which is vital for patient care in elder care and rehabilitation contexts.

### 4.1. Neurological System Diagnosis

The application of TENGs in neurological system diagnosis marks a significant evolution in medical technology, particularly in PoCD for neurological conditions such as Parkinson’s disease.

Kim et al. focused on creating a highly adaptable and biocompatible TENG using natural biomaterials [70] (Figure 4a). Their hydrogel TENG incorporates a unique blend of catechol, chitosan, and diatom to form the hydrogel electrode. The electrode is complemented by an M-shaped Kapton film. This design ensures the device is not only stretchable and self-healing but also sensitive to low-frequency motions, which are crucial indicators in the diagnosis and monitoring of Parkinson’s disease. The ability to capture such subtle movements allows for a detailed understanding of the disease’s progression and the effectiveness of ongoing treatments. The natural composition of the hydrogel also ensures biocompatibility, reducing the risk of adverse reactions when the device is used for extended periods.

On the other hand, Yuce et al. developed a custom-made TENG that is tailored for the specific needs of Parkinson’s disease patients [78]. Their device leverages a combination of a dielectric and aluminum electrodes to create a flexible and sensitive tool for monitoring minute muscular activities and hand movements. This sensitivity is vital for accurately assessing the progression of Parkinson’s disease. The flexible nature of the device ensures that it can be comfortably worn by patients, facilitating long-term monitoring without causing discomfort or hindering daily activities. By tracking these subtle movements, the device provides valuable data that can be used to tailor individual treatment plans and monitor the effectiveness of various therapeutic interventions.

The significance of these TENG-based systems in PoCD lies in their ability to facilitate early detection and continuous monitoring of Parkinson’s disease. By providing real-time data on tremors and muscular movements, these devices aid in the timely adjustment of treatment plans and can potentially slow the progression of the disease.

In another effort to advance PoCD in neurology, Wang et al. have developed a pioneering device comprising two core modules: a flexible strain sensor fabricated from graphene oxide-polyacrylamide (GO-PAM) hydrogels, and a sophisticated data processing module [71] (Figure 4b). This device is specifically tailored for clinical settings, targeting the diagnosis of Parkinson’s disease and hemiplegia. The innovative self-powered strain sensor, utilizing the sensitivity of GO-PAM hydrogels, is highly effective in detecting subtle human movements such as variations in gait patterns. This in-shoe wearable monitoring system, which employs an artificial neural network algorithm, demonstrates remarkable accuracy in differentiating between normal and pathological gait patterns; this is a crucial aspect in the daily-life assessment of patients. Its utility extends beyond mere diagnosis, and it plays a vital role in rehabilitation evaluation and treatment planning for patients. This device exemplifies the potential of PoCD in transforming the management of neurological conditions, offering a practical, non-intrusive solution for early diagnosis, continuous monitoring, and effective treatment of diseases such as Parkinson’s.

The integration of TENG technology in the diagnosis of neurological disorders such as Parkinson’s not only improves the accuracy of diagnosis but also enhances the quality of life of patients by enabling better disease management. Moreover, the wearable nature of these sensors makes them ideal for continuous monitoring, which is a critical factor in managing chronic neurological conditions.

### 4.2. Musculoskeletal System Diagnosis

In the realm of musculoskeletal system diagnostics, the versatility of TENG technology comes to the forefront. Musculoskeletal disorders, encompassing a range of conditions from osteoporosis to muscular dystrophies, demand precise and early diagnosis for effective management. TENG devices, through their ability to detect and measure minute mechanical deformations and movements, offer non-invasive monitoring of joint movements, muscle strength, and bone health. TENG technology is capable of not only enhancing diagnostic accuracy but also revolutionizing patient care in orthopedics and rehabilitation.

#### 4.2.1. Motion Diagnosis

In the dynamic field of musculoskeletal system diagnostics, the work of Shi et al. stands out as a notable contribution [72] (Figure 4c). They developed a biosensor that ingeniously integrates conductive polymers within a TENG framework. This biosensor distinguishes itself through its enhanced performance capabilities, particularly in the high-precision detection of subtle finger movements. Such a feature is invaluable in assessing conditions that impair fine motor skills, making it a critical tool in the realm of point-of-care diagnostics (PoCD). Furthermore, the sensor’s compatibility with wireless technology marks a significant advancement. It enables remote monitoring of patients’ finger movements, providing crucial real-time data for diagnostic and treatment procedures from a distance. This innovative approach aligns perfectly with the objectives of PoCD, offering immediate and accurate patient assessment in various settings.

Similarly, the work of Zhang et al. represents a significant stride in the field of musculoskeletal diagnostics [73] (Figure 4d). They have developed a lightweight, wearable TENG device equipped with fall detection capabilities, which is a vital feature for the elderly and individuals with specific health conditions. The key innovation of this device lies in its incorporation of three-dimensional polypyrrole nanoarrays, which enable the device to accurately differentiate between normal activities and fall events. This functionality is particularly crucial in proactive fall prevention strategies. Upon detecting a fall, the device can promptly trigger alerts for assistance, playing a pivotal role in rapid emergency response and injury prevention. This technology emerges as a cornerstone in PoCD, enhancing safety and monitoring for vulnerable groups and contributing significantly to injury prevention and emergency management strategies.

Building on the innovative advancements in the field of motion diagnosis, the work of yang et al. [74] represents a significant leap forward; their transparent self-powered triboelectric sensor is based on PVA/PA hydrogel. This system, ingeniously crafted for medical nursing human–machine interfaces (HMI), is a testament to the intricate integration of TENG technology in healthcare (Figure 4e). The core of this system lies in its tripartite modular design, which encompasses a TENG device that is adept at converting mechanical stimuli into electrical signals; a sophisticated multichannel signal processing unit; and an intuitive user interface module. The TENG device, a marvel of engineering, is strategically designed to be attached to various body joints, making it highly versatile. Its capability to translate even the subtlest of mechanical movements into discernible electrical signals is groundbreaking. The signal processing unit, with its multichannel architecture, meticulously collects, processes, and encodes these signals, ensuring their accuracy and reliability. The user interface module then emerges as the crucial link, providing real-time feedback and assistance, thereby bridging the gap between patients and medical staff in a seamless manner.

Further extending the application spectrum of TENG technology in the realm of healthcare is the innovative work of Yuan et al. [75]. Their proposition of a flexible sliding sensor for simultaneous monitoring deformation and displacement on a robotic hand/arm is a testament to the multifunctional capabilities of TENGs. This system ingeniously combines a capacitive sensor and a TENG-based sensor in a cohesive unit (Figure 4f). The capacitive sensor is meticulously engineered to detect compressive stress and deformation, providing valuable data on the structural integrity and stress distribution. Concurrently, the TENG-based sensor adeptly identifies displacement and velocity, offering insights into the dynamics of motion. The integration of these sensors in a modular design is a stroke of genius, allowing the device to capture a diverse range of physical changes through nuanced electrical signals. This not only facilitates precise control in applications such as robotic gripping but also aids in the prevention of unintended slippage or damage during these processes. The inherent adaptability of this technology, underscored by its self-powering capabilities, makes it a versatile tool that is suitable for a wide array of applications. From robotic hands and arms to advanced touch screens and electronic skins, the potential of this technology is vast and varied.

#### 4.2.2. Bone Morphology Diagnosis

Bone morphology diagnosis represents an important aspect of musculoskeletal healthcare. The application of TENGs in this field has opened new avenues for the non-invasive and accurate assessment of bone health. TENG technology, as applied in bone morphology diagnostics, utilizes the principle of converting mechanical energy into electrical energy. This conversion enables the detailed analysis of bone structure and composition without the need for invasive procedures. Sensitivity and accuracy are vital for detecting subtle changes in bone morphology, which are often indicators of conditions such as osteoporosis or other bone degenerative diseases.

Li et al.’s innovative approach in the field of clinical applications involves the development of a self-powered wireless sensor network, which leverages the capabilities of TENG technology [79]. In addition to energy harvesting, power management modules, and wireless communication, Li et al. have developed a high-sensitivity stretch sensing device that is capable of detecting subtle joint and spinal motions with remarkable precision and robustness. Its high sensitivity allows for the monitoring of even minute movements, making it exceptionally suitable for applications such as spinal monitoring. The precision of this sensor is instrumental in identifying and mitigating risks associated with long-term posture-related diseases, which is a common concern in clinical and rehabilitative settings.

Liu et al. have made a notable breakthrough in the field of orthopedic diagnostics by developing a self-powered, modular sensor for artificial joint wear debris, employing TENG technology (Figure 4g) [76]. This sensor adeptly transforms mechanical energy from joint movements into electrical signals, facilitating the timely and onsite identification of wear debris in artificial joint replacements. Representing a significant stride in the realm of biomedical sensors, this TENG-based sensor broadens the scope of TENG technology in intelligent healthcare, enhancing the diagnosis and management of artificial joint health.

In another significant advancement, Kaveh Barri and colleagues have engineered an innovative series of intelligent medical implants equipped with a dual-module system: a TENG featuring an auxetic microstructure and a wireless communication module; these are shown in (Figure 4h) [77]. These advanced self-aware metamaterial implants are designed to monitor bone healing processes and evaluate different stages of spinal fusion, performing continuous measurements of stability and load distribution without the need for radiographic imaging. Its development underscores the critical importance of modular design in the real-time diagnosis and monitoring of bone morphology, setting a new standard in the field of orthopedic healthcare.

The future of bone morphology diagnosis using TENG technology looks promising. With ongoing advancements, these devices could become more compact, user-friendly, and integrated with other digital health tools. This integration could lead to comprehensive health monitoring systems that not only assess bone health but also provide insights into overall musculoskeletal wellbeing. Additionally, there is potential for these technologies to be used in remote settings, expanding access to bone health diagnostics in underserved areas.

## 5. TENG for Sweat and Bacteria Real-Time Diagnosis

### 5.1. Sweat Sensing

In the rapidly evolving field of PoCD, the monitoring of biomarkers components in sweat has been revolutionized by the advent of TENG technology. This advancement is showcased in the development of innovative, flexible nanosensors that are capable of non-invasive creatinine detection and real-time sweat composition analysis; this marks a significant stride in health monitoring (Table 4).

#### 5.1.1. Lactic Acid and Creatinine Detection

The innovative self-powered molecular imprinted polymers-based triboelectric sensor (MIP-TES), designed by Kanokpaka et al., represents a significant advancement in wearable sweat sensors, particularly for lactate detection [80] (Figure 5a). This sensor uniquely addresses the challenges previously faced in this field, such as the use of noble materials, dependence on immobile power supplies, and complexities in circuit connections. The MIP-TES offers a multifunctional, non-invasive approach for specific and simultaneous detection of lactate in human sweat. Its functioning is based on a free-standing PVDF/graphene flexible electrode modified with poly(3-aminophenyl boronic acid), which undergoes surface property changes upon lactate adsorption, enabling selective lactate sensing. The integration of this sensor with a triboelectric nanogenerator system allows it to harvest mechanical energy from human movement, converting it into electrical output. This feature is particularly noteworthy as it enables the sensor to power LED lights directly, without external energy sources, indicating lactate concentration levels.

**Figure 5 sensors-24-00511-f005:**
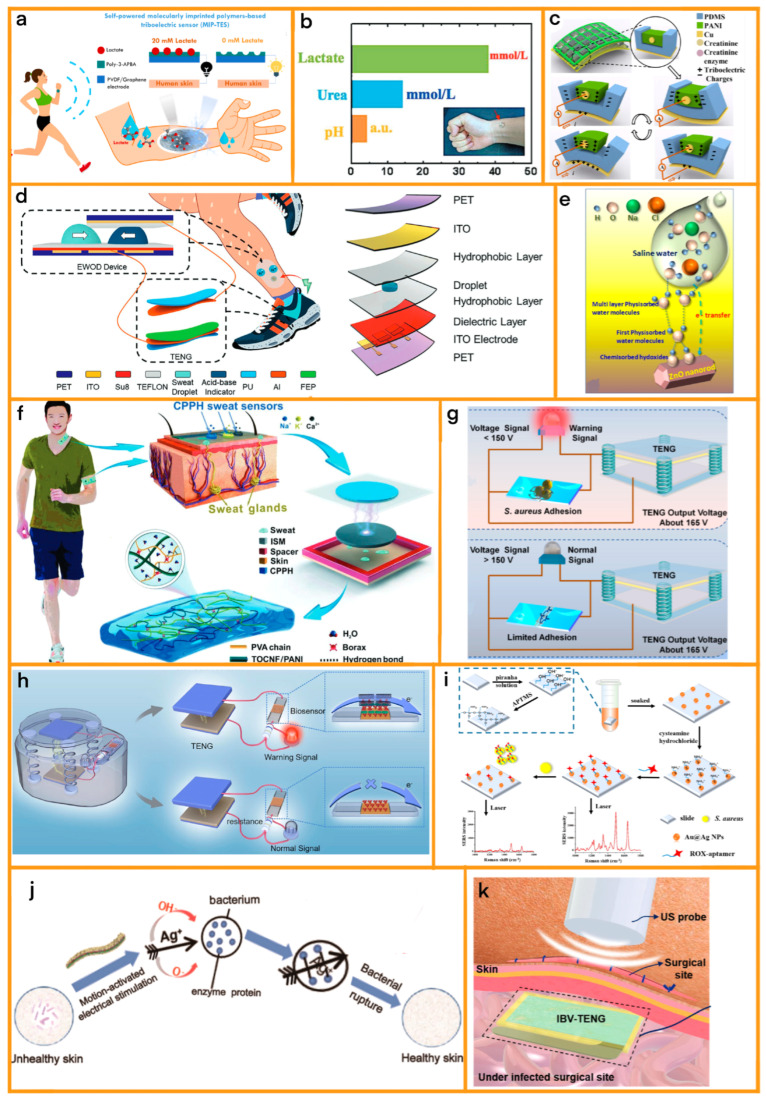
(**a**) Self-powered molecularly imprinted polymers-based triboelectric sensor (MIP-TES) for non-invasive lactate monitoring in human sweat [Source: Elsevier, 2022] [80]. (**b**) Lactate and urea concentrations detected in sweat during exercise (running) using artificial skin. Inset: Optical image of the artificial skin attached to a human wrist during a test, demonstrating the device’s practical application and wearability. [Source: Advanced Materials, 2019] [81]. (**c**) Working mechanism of the triboelectric nanogenerator-based nanosensors. [Source: Elsevier, 2021] [82]. (**d**) Schematic illustration of a WSMP worn on the leg to detect the pH level of sweat. It contains an electrowetting on dielectrics (EWOD) device and a triboelectric nanogenerator (TENG) to collect and transport sweat droplets by dielectric wetting effect and eventually merge and react with a pH indicator [Source: AFM, 2022] [83]. (**e**) Ongoing mechanism of sweat sensing from the event of adsorption of saline water on ZnO surface [Source: Elsevier, 2023] [84]. (**f**) Cellulose-based conductive hydrogel used for self-powered sweat sensing. [Source: Advance materials, 2022] [85]. (**g**) Schematic illustration of S. aureus (Staphylococcus aureus) detection based on a self-powered biosensing system. The diagram shows the S. aureus detection process in a liquid environment and a vertical contact-separation TENG as the voltage signal source. [Source: Elsevier, 2022] [86]. (**h**) Schematic illustration of bacterial detection process. When there are bacteria, the alarm will be triggered; when there are no or few bacteria, the alarm will not be triggered [Source: Elsevier, 2022] [87]. (**i**) Schematic illustration of the developed detection method, detailing the method’s design and approach to detection. [Source: Elsevier, 2023] [88]. (**j**) Skin health restoration process of the wearable self-powered long-acting germicidal devices [Source: ACS, 2023] [89]. (**k**) Schematic illustration of IBV-TENG based on US under the surgical site to prevent SSI by ES [Source: Advance science, 2023] [90].

**Table 4 sensors-24-00511-t004:** TENG in sweat and bacteria real-time diagnosis.

Date	Sizes	Energy Sources	Performance	Applications
2022 [80]	None	Movement	38 μW	Lactic acid detection
2019 [81]	2 cm × 2 cm	Movement	0.8 μA	Lactic acid detection
2021 [82]	2 cm × 4 cm	Movement	0.32 nW	Creatinine detection
2022 [83]	None	Movement	60 V	Sweat electrolyte analysis
2023 [84]	2.5 cm × 2.5 cm	Movement	0.02 V/μL	Sweat electrolyte analysis
2022 [85]	1 cm × 1 cm	Movement	0.069 mmol^−1^ (Sensitive)	Sweat electrolyte analysis
2022 [86]	3.5 cm × 3.5 cm	Vibration	165 V	Bacteria diagnosis
2022 [87]	None	Vibration	160 V	Bacteria diagnosis
2023 [89]	None	Vibration	1080 V	Sterilization
2023 [90]	1 cm × 2 cm	Vibration	22 μA	Sterilization

A trailblazing advancement in this arena has been achieved by Gao et al. [81], who have harnessed the innovative capabilities of TENGs to devise a wearable device adept at concurrently monitoring biochemical and electrophysiological signals from human sweat. This device represents a significant leap forward in non-invasive health monitoring, ingeniously amalgamating multiple sensing modalities into a singular, wearable platform (Figure 5b). It features a flexible lactate sensor, utilizing an enzyme-based reaction to translate the concentration of lactate in sweat into a proportional electrical signal. This development is pivotal, as fluctuations in lactic acid levels can be indicative of a range of physical states and athletic performance metrics. Enhancing the functionality of the lactate sensor is a wireless potentiostat, which adeptly amplifies and digitizes the signal, thus ensuring accurate, real-time data transmission. In addition, the device incorporates a flexible electrocardiogram (ECG) sensor, which skillfully captures heart electrical activity directly from the skin, thereby providing invaluable insights into cardiac health.

The flexible creatinine nanosensor created by Luo et al. [82] is a notable example of this technology. This sensor uses a combination of polyaniline (PANI) and polydimethylsiloxane (PDMS) (Figure 5c). The sensor operates based on the synergy between the TENG and creatinine enzymatic reaction. The enzyme-modified TENG, comprising PANI and PDMS, exhibits changes in electroconductivity in response to enzymatic reactions. These changes directly affect the triboelectric output, allowing the sensor to accurately measure ambient creatinine levels. This is particularly crucial in renal health management, where monitoring creatinine levels is vital. The sensor shows remarkable sensitivity at room temperature and impressive selectivity against NaCl, glucose, and urea. It also supports a wide range of flexibility in bending angle measurements, making it suitable for wearable sensing applications.

#### 5.1.2. Sweat Electrolyte Analysis

In the realm of wearable health monitoring, Shen et al. have designed an innovative device, known as the Wearable Sweat Monitoring Platform (WSMP), which harnesses the combined capabilities of an Electrowetting on Dielectrics (EWOD) device and a triboelectric nanogenerator (TENG) [83] (Figure 5d). This platform is a groundbreaking advancement in sweat analysis, which efficiently analyzes the complex composition of solutes and metabolites in human sweat, thereby providing real-time insights into the wearer’s health status. The WSMP is adept at actively controlling the motion of sweat droplets, enabling precise collection and transport of these droplets across various analysis chambers. This movement is facilitated by the dielectric wetting effect, which allows the droplets to merge and interact with pH indicators, providing valuable health data.

In their seminal work, Baro et al. developed a cutting-edge, textile-based, wearable sweat sensor utilizing the principles of triboelectrification, embodied in a single electrode triboelectric nanogenerator (STENG) [84] (Figure 5e). This innovation is marked by the integration of chemically synthesized zinc oxide (ZnO) nanorods on a textile substrate, enabling dual functionality as both a motion sensor activated by biomechanical movements, and a sweat sensor that is sensitive to variations in saline water concentration. The STENG’s efficacy is heightened by the attachment of hydrated chloride ions from saline water to physisorbed water molecules on ZnO, enhancing the electron count in ZnO’s conduction band, thus boosting charge transfer and output voltage. A miniaturized prototype of this STENG, approximately 1 cm in diameter, demonstrates remarkable efficiency in detecting sweat when attached to the human body; it has a sensitivity of about 0.02 V/μL and a detection limit near 4.8 μL. Notably, when this prototype is integrated into a shoe insole it can detect sweat during foot movement. The study also explores the STENG’s optimization, particularly its active triboelectric layer, with ZnO nanorods strategically placed on a cotton base and various materials such as aluminum, nitrile, PET, and PTFE being tested for the counter triboelectric layer. The choice of PTFE-ZnO stands out for generating superior output voltage. This refined STENG exhibits an amplified output in contact with saline water, which is linked to an increase in n-type conductivity in ZnO.

Similarly, the groundbreaking sweat sensor designed by Qin et al. [85] integrates PANI into a hydrogel electrode (Figure 5f). This sensor facilitates the real-time analysis and wireless transmission of vital electrolytes, such as sodium (Na^+^), potassium (K^+^), and calcium (Ca^2+^), which are found in sweat. The sensor’s design includes a cutting-edge integration of PANI with cellulose nanocomposites to create a hydrogel electrode, which exhibits remarkable tensile and electrical self-healing efficiency, stretchability, and conductivity. This sensor’s extraordinary flexibility, stability, sensitivity, and selectivity make it an invaluable tool for health monitoring, with applications ranging from sports to clinical settings. It can detect Na^+^, K^+^, and Ca^2+^ ions with high sensitivity and wirelessly transmit data to a user interface, providing easy access to important health information.

### 5.2. Bacteria Diagnosis and Sterilization

#### 5.2.1. Bacteria Diagnosis

Advancing further into the realm of PoCD, we explore the multifaceted applications of TENG technology in sweat composition analysis and bacterial detection. These developments represent remarkable achievements in health monitoring and disease diagnostics, showcasing the versatility of TENGs in improving healthcare outcomes.

In the field of PoCD, especially in the detection of bacterial infections caused by Gram-positive bacteria, the introduction of TENG technology has brought about significant advancements. A particularly noteworthy development is the research by Wang et al., who engineered a TENG-based biosensing system that is adept at detecting Gram-positive bacteria in liquid mediums [86]. This system is ingeniously designed, featuring a blend of polyamine and vancomycin, skillfully immobilized on the surface of indium tin oxide (ITO) glass (Figure 5g). It employs guanidine-functionalized multi-wall carbon nanotubes (CNT Arg) as potent signal amplification materials. The distinctive aspect of this system lies in its ability to specifically target and identify Gram-positive bacteria in a solution through measuring voltage changes in the biosensor. This innovation not only signifies a leap forward in clinical diagnostics of bacterial infections but also showcases its potential for environmental monitoring, underscoring its versatility and practical applicability.

In a significant advancement for marine construction and environmental monitoring, Zhou et al. have developed a novel self-powered microbiosensor system using a triboelectric nanogenerator (TENG) to detect Gram-negative bacteria (Gnb), such as sulfate-reducing bacteria (SRB), which are primary agents of microbiologically induced corrosion (MIC) [87] (Figure 5h). This innovative system combines concanavalin A (ConA)-modified indium tin oxide glass (ITO-ConA) and ConA-functionalized carboxyl multiwalled carbon nanotubes (CNT-ConA), which together enhance detection sensitivity and specificity. The ITO-ConA component captures Gnb through specific interactions between ConA and the lipopolysaccharide of Gnb, while the high-conductivity CNT-ConA serves as a signal amplifier by modifying the resistance in response to varying Gnb concentrations. The incorporation of TENG ensures a stable and independent power supply, eliminating the need for external power sources. The system’s effectiveness is evidenced by its ability to detect *Desulforibrio* sp. Huiquan2017, a typical SRB, with a linear relationship between peak voltage and bacterial concentration across a broad detection range. Additionally, an integrated alarm circuit in the microbiosensor allows immediate visual indication through a light-emitting diode. This groundbreaking development not only simplifies operations compared to traditional methods but also minimizes environmental impact and dependence on complex instruments.

In a concurrent breakthrough, Ma and his team have made substantial contributions in this arena with their novel Surface-Enhanced Raman Spectroscopy (SERS) method, which is adept at detecting Staphylococcus aureus, a prevalent Gram-positive bacterium (Figure 5i) [88]. Their approach involves the use of Au@Ag nanoparticles/slide as an enhanced substrate in an aptamer-based SERS methodology. This strategy facilitates non-invasive and efficient detection of Staphylococcus aureus, playing a crucial role in diagnosing and managing a spectrum of diseases including respiratory infections, bronchitis, pneumonia, skin conditions, and surgical incision infections. The method’s design is pivotal in enabling early detection and treatment, potentially curtailing the severity of complications and optimizing the use of medical resources.

In summary, these innovative applications of TENG technology in bacterial diagnostics have not only taken significant strides in the clinical diagnosis of bacterial infections but also demonstrated their multifunctionality and practicality in environmental monitoring. These advancements not only show potential in optimizing the use of medical resources but also pave the way for accurate and timely diagnosis of bacterial infections.

#### 5.2.2. Sterilization

The concept of a self-powered sterilization system is a response to the limitations of traditional sterilization methods, which often require external power sources or chemicals. These newer technologies leverage the principles of TENGs, which convert mechanical energy into electricity, thus providing a sustainable and portable solution for sterilization. This is particularly relevant in the context of wearable electronics, where prolonged contact with the skin can lead to bacterial growth and skin infections.

In the realm of wearable electronics, Lei et al. have pioneered a groundbreaking, self-powered sterilization system designed to address the growing concern of skin infections caused by prolonged wearing of electronic devices [89] (Figure 5j). This innovative solution hinges on a nano/microporous fiber triboelectric nanogenerator (NMF-TENG) that serves as a sustainable energy source, coupled with an interdigital electrode enhanced by silver nanowires (AgNWs) and carbon nanotubes (CNT) for superior bactericidal efficacy. The unique design of the indium tin oxide (ITO)-based electrode significantly amplifies the local electric field to an impressive 1 MV/m, which is potent enough to eradicate bacteria. This system boasts an impressive sterilization rate of up to 90%, underscoring its potential to revolutionize the sterilization aspect of wearable technology. The creation process involves a meticulous two-component electrospinning technique, yielding a robust and flexible PA66/EC nanofiber composite film that enhances the frictional contact area of the NMF-TENG, thereby generating a higher surface charge density. This results in an impressive output of 302 V and 0.4 μA at a low-frequency motion of 1 Hz, ensuring a reliable power supply for the device. Furthermore, the patterned AgNWs/CNT/ITO nanowire interdigital electrode, which is fabricated through spin coating and direct laser writing, exhibits commendable flexibility and conductivity, seamlessly conforming to the human body’s contours. The device leverages the “tipeffect” of the nanowires to produce a sustained 1 MV/m electric field, which not only disrupts the structural integrity of microorganisms but also facilitates the release of Ag+ ions, which are known for their antibacterial properties.

Imani et al. have introduced a groundbreaking solution to combat surgical site infections (SSI) with their ultrasound (US)-driven implantable, biodegradable, and vibrant triboelectric nanogenerator (IBV-TENG) [90] (Figure 5k). This novel device addresses the significant challenge of antimicrobial resistance by providing a method to eliminate microorganisms in deep tissues through electrical stimulation, addressing a key concern in global health. The IBV-TENG, upon activation by ultrasound, generates approximately 4 V and 22 μA, demonstrating its efficacy in vitro by inactivating nearly 100% of Staphylococcus aureus and about 99% of Escherichia coli. Ex vivo studies further validate its ability to inactivate bacteria under porcine tissue. A critical feature of this technology is its biodegradability, which allows for on-demand dissolution controlled by ultrasound intensity, thus eliminating the need for surgical removal. This innovation in antibacterial technology not only offers a potent countermeasure against SSIs but also enhances life expectancy and healthcare quality by preventing deep tissue microorganism growth. The use of electrical stimulation (ES) by the IBV-TENG, a known method for inhibiting microorganisms in infected wounds, marks a significant advancement in the prevention and treatment of SSIs especially against antimicrobial-resistant strains, showcasing the potential of implantable TENGs in medical

## 6. Conclusions and Perspectives

After a comprehensive review of Triboelectric Nanogenerators (TENGs) in the context of point-of-care diagnostics (PoCD), we recognize the potential of this technology in enhancing medical diagnostics, while also acknowledging a range of challenges it faces. Firstly, the choice and durability of materials are major limiting factors in the application of TENGs. The materials currently used may encounter issues such as high costs, sourcing difficulties, or environmental impacts. Additionally, the variability in environmental conditions such as temperature, humidity, and atmospheric pressure could lead to inconsistent energy outputs from TENGs. The wear and tear of materials due to continuous friction also affects the efficiency and lifespan of TENGs. Despite the challenges, the prospects for TENGs in the field of medical diagnostics remain promising. Future research should focus on addressing these challenges, to harness the significant potential of TENGs to provide sustainable and efficient energy solutions, especially in resource-limited settings.

### 6.1. TENG’s Energy Collection in PoCD

In the realm of PoCD, the integration of TENGs plays a transformative role. TENGs, by leveraging energy from ambient sources, provide a sustainable and efficient power supply for a variety of portable diagnostic devices. This is particularly valuable in PoCD, where the need for rapid, reliable, and accurate diagnostic tools is paramount. The adaptability and efficiency of TENGs make them ideal for these settings, offering a power solution that aligns with the dynamic and diverse requirements of modern healthcare environments. Their ability to operate in various conditions without compromising on power efficiency makes TENGs a revolutionary addition to PoCD technologies, enabling more streamlined and effective patient care.

#### 6.1.1. Increased Output Power for PoCD

Enhancing the output power of TENGs is crucial for their effective application in PoCD. This enhancement can be achieved by meticulously modifying the surface of the electrode layer. Employing a combination of chemical and physical methods, such as grafting chemical groups or altering the surface texture through techniques such as laser engraving or plasma processing, can significantly boost electron transmission efficiency. Concurrently, the choice of biocompatible materials for the friction layer is vital. These materials should not only possess a high surface charge but also exhibit higher friction electronegativity, all while maintaining compatibility with biological systems. Furthermore, the overall design and structure of TENGs are tailored to integrate seamlessly with medical devices, ensuring that they align with the practical and ergonomic needs of PoCD. This involves designing TENG structures that conform to the human body, enhancing comfort for patients during prolonged use and ensuring that these devices are in harmony with the physical and operational demands of PoCD environments. These advancements are aimed at producing TENGs that are not just more powerful, but also more practical and user-friendly for long-term application in diverse clinical settings.

#### 6.1.2. Stability in TENG for PoCD

Stability is a critical factor in the application of TENG technology for PoCD. In PoCD, a consistent and reliable power supply is paramount, particularly for chronic disease management. TENGs, which generate electricity through continuous contact between layers, often experience significant wear in the friction layer. To mitigate this and enhance the service life and stability of TENGs, future developments should focus on identifying wear-resistant materials and exploring new friction methods with lower loss rates. Additionally, considering the integration of TENGs in the human internal environment, material choices should prioritize acid and alkali resistance and incorporate waterproof coatings to mitigate the effects of humidity. For medical diagnostic applications, the stability of TENGs can be further improved by integrating filter circuits to reduce interference from minor physiological fluctuations, ensuring consistent and accurate diagnostic results.

### 6.2. TENG for Diagnosis in PoCD

TENG technology in PoCD offers a paradigm shift in medical diagnostics. Its application in PoCD is characterized by enhanced sensitivity, wireless data transmission, and comfort, aligning with the requirements of modern medical practices. TENG’s adaptability and efficiency make it a valuable tool in developing diagnostic devices that are not only precise but also patient-friendly, facilitating early diagnosis and ongoing monitoring in various clinical and non-clinical environments.

#### 6.2.1. Wireless Data Transmission

In PoCD, wireless data transmission is essential for continuous monitoring and real-time data analysis. Traditional wired systems are often impractical due to their bulkiness and constraints. TENG technology can revolutionize this aspect by enabling more energy-efficient wireless communication. The potential use of TENG’s electrostatic induction effect to drive signal transmission presents a novel approach, although its efficacy is currently limited by TENG’s internal resistance. Future research should focus on enhancing this aspect, thereby making TENG an even more powerful tool for wireless communication in PoCD devices.

#### 6.2.2. Enhanced Sensitivity in TENG for PoCD

The enhancement of sensitivity in TENG technology is of key importance for the advancement of PoCD. With the evolving complexity of medical diagnostics, the need for devices that can detect subtle physiological changes is increasingly important. Improving TENG’s sensitivity involves the development of materials that can respond to minute changes, coupled with sophisticated filter circuits that minimize environmental noise and interference. This advancement will enable more precise and accurate diagnostics, which are essential for early disease detection and management. The use of high-precision fabrication methods, such as 3D printing or laser etching, could further refine device structure, reducing systemic errors and increasing diagnostic accuracy.

#### 6.2.3. Comfort and Wearability in TENG-Based PoCD Devices

Comfort and wearability are paramount in the design of TENG-based PoCD devices, especially for those intended for long-term wear or continuous monitoring. Future developments should focus on creating devices that are not only functionally effective but also comfortable for patients. This involves exploring innovative materials that are lightweight, breathable, and skin-friendly, such as natural fibers, to enhance patient comfort. Ergonomic design is equally important, ensuring that devices conform to the body’s contours and movements without causing discomfort. Personalized design, taking into account individual patients’ needs and preferences, will also play a crucial role in improving the user experience and acceptance of these devices in clinical and everyday settings.

## Figures and Tables

**Figure 1 sensors-24-00511-f001:**
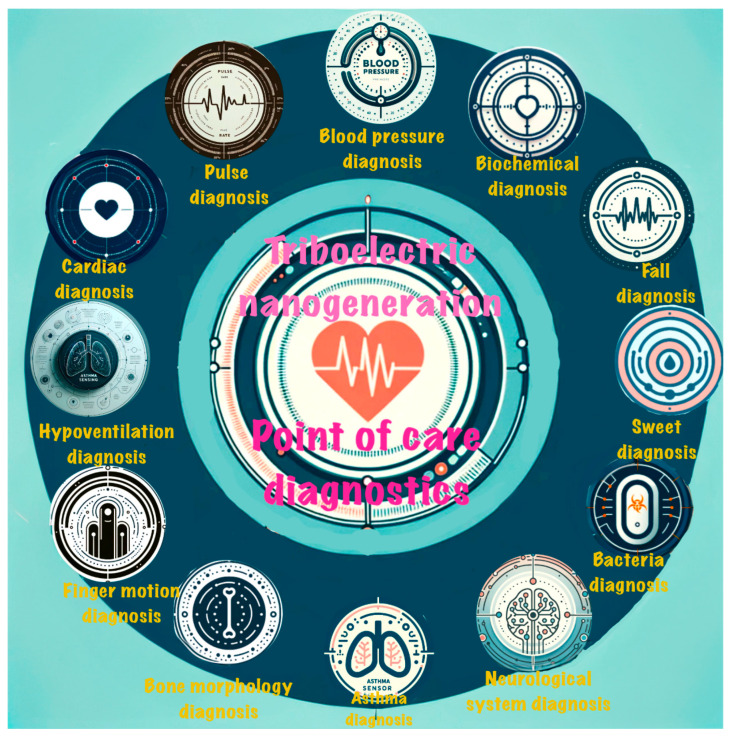
TENG-based point-of-care diagnostics.

**Figure 2 sensors-24-00511-f002:**
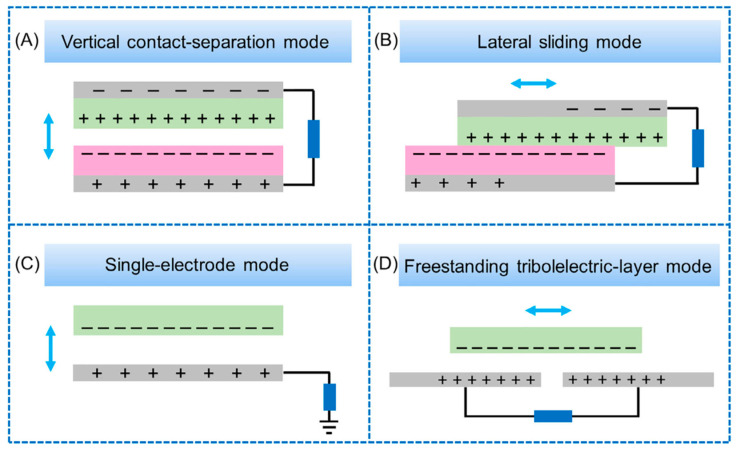
(**A**) Vertical contact-separation mode. (**B**) Lateral sliding mode. (**C**) Single-electrode mode. (**D**) Freestanding triboelectric-layer mode [45]. Zhao et al. (2020), John Wiley and Sons.

**Figure 3 sensors-24-00511-f003:**
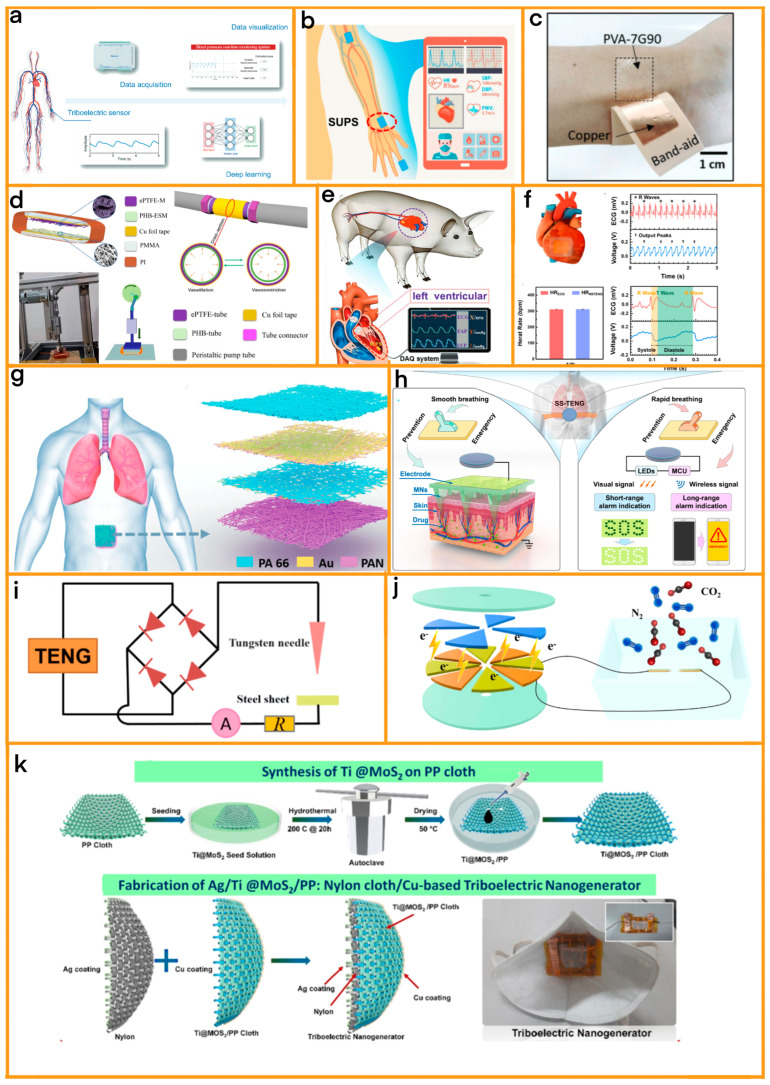
(**a**) Schematic diagram of a blood pressure testing device, illustrating its various components and how they function to measure blood pressure. [Source: Nano Research, 2022] [55]. (**b**) Illustration of the integration of SUPSs for non-invasive multi-indicators cardiovascular monitoring, depicting how the system non-invasively monitors various cardiovascular indicators. [Source: Elsevier, 2021] [56]. (**c**) The optical image of the PVA-7G90 film attached to the human wrist, showing the film in a practical, real-life application, highlighting its wearability and use case. [Source: Advanced Materials, 2020] [57]. (**d**) Schematic diagram of the ePTFE/PHB TENG structure. [Source: Elsevier, 2023] [58]. (**e**) Schematic diagram of the semaphore acquisition from the SEPS (Specific Electronic Processing System) implanted into an adult Yorkshire swine’s heart. [Source: Advanced Materials, 2019] [59]. (**f**) Schematic illustration showing the NSTENG (Nanostructured Triboelectric Energy Generator) implanted at the anterior wall (AW) of the heart. The measured Voc (Voltage open circuit) of the NSTENG at AW and the corresponding ECG (electrocardiogram). The heart rate measured by the NSTENG (HRNSTENG) at AW in comparison with that measured by ECG (HRECG). Magnified Voc at AW and the corresponding ECG. [Source: Elsevier, 2021] [60]. (**g**) Application scenario of the SANES (Surface Acoustic Nanoelectrical Sensors) attached on the surface of the abdomen for respiratory monitoring. Schematic illustration of the SANES. Enlarged view of the layers of Au (gold) electrodes coated on the surface of PAN (Polyacrylonitrile) nanofiber film and PA 66 (Polyamide 66) nanofiber film. [Source: Advanced Materials, 2021] [62]. (**h**) Prevention function of SS-TENG (Self-Sufficient TENG) is activated by smooth breathing motion. A triboelectrically driven transdermal drug delivery system with the dissolvable microneedles (MNs) array still operating. Emergency function of SS-TENG is self-switched by rapid breathing motion. Visual and wireless alarm modules transfer emergency signals. [Source: Elsevier, 2023] [63]. (**i**) Circuit diagram of the self-powered CO_2_ sensor [Source: Elsevier, 2018] [64]. (**j**) Structure and schematic diagram of a self-powered CO_2_ gas detection system [Source: Elsevier, 2023] [65]. (**k**) Synthesis of Ti@MoS_2_ on PP Cloth and Steps for the fabrication of Ag/Ti@MoS_2_:PP/Nylon/Cu Triboelectric Nanogenerator and real-time representation of triboelectric mask nanogenerator [Source: Elsevier, 2023] [66].

**Figure 4 sensors-24-00511-f004:**
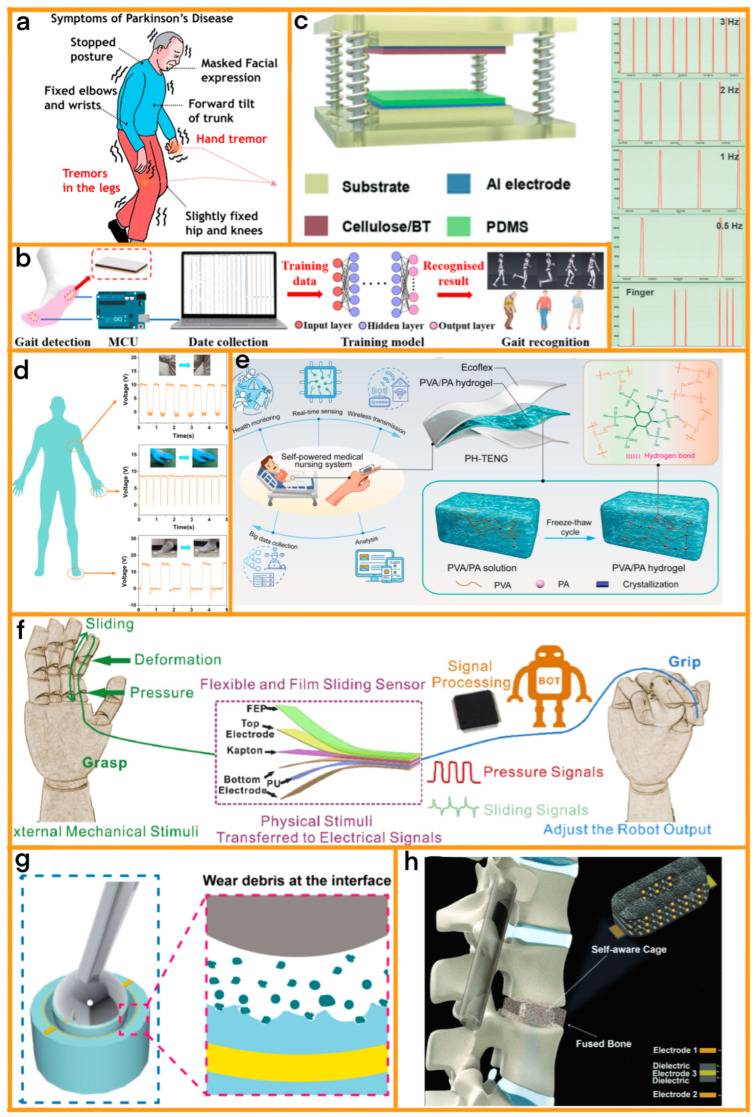
(**a**) Typical symptoms of Parkinson’s disease, depicting common physical and neurological signs associated with the disease. [Source: Elsevier, 2021] [70]. (**b**) The process of human gait recognition by a gait recognition model, illustrating how human walking patterns are analyzed and recognized. [Source: Elsevier, 2022] [71]. (**c**) The structure of the TENG based on the vertical contacting–separating operation mode, showing the design and mechanism of this energy-harvesting device, and the real-time display of the machine operation and finger movement on the cellphone screen. [Source: Advanced Materials, 2020] [72]. (**d**) The voltage output when PPy-PVDF TENG is worn on different parts of the human body such as underarms, hands, and feet. [Source: ACS (American Chemical Society), 2022] [73]. (**e**) Application and architecture of the PH-TENG and microstructure of the PVA/PA hydrogel [Source: Elsevier, 2022] [74]. (**f**) Schematic structure and working principle of the flexible sliding sensor [Source: Elsevier, 2020] [75]. (**g**) Schematic illustration of the wear debris sensor, outlining the design and function of a sensor that detects wear particles. [Source: Elsevier, 2021] [76]. (**h**) A multifunctional nanogenerator interbody fusion cage with self-recovering, self-sensing, and energy-harvesting functionalities, implanted during spinal fusion surgery; this highlights its multifaceted applications in medical procedures. [Source: Advanced Materials, 2022] [77].

**Table 1 sensors-24-00511-t001:** TENG in cardiovascular diagnosis.

Date	Sizes	Energy Sources	Performance	Applications
2022 [55]	5 cm × 5 cm	Vibration	0.89 V/kPa	Blood pressure diagnosis
2021 [56]	1 cm × 2 cm	Vibration	10.29 nA/kPa	Pulse diagnosis
2020 [57]	1.5 cm × 2 cm	Vibration	5.9 V	Pulse diagnosis
2023 [58]	5 cm × 5 cm	Vibration	19.5 mA/m^2^	Pulse diagnosis
2019 [59]	1 cm × 1.5 cm × 0.1 cm	Vibration	1.195 mV/mmHg	Cardiac diagnosis
2021 [60]	2 cm × 2 cm	Vibration	51.74 nA	Cardiac diagnosis

**Table 2 sensors-24-00511-t002:** TENG in respiratory system diagnosis.

Date	Sizes	Energy Sources	Performance	Applications
2021 [62]	4 cm × 4 cm	Vibration	330 mW m^−2^	Hypoventilation syndrome diagnosis
2023 [63]	None	Vibration	214 μW cm^−2^	Asthma diagnosis
2018 [64]	None	Rotation	20.1 μA	Carbon dioxide sensing
2023 [65]	None	Rotation	500 V	Carbon dioxide sensing
2023 [66]	None	Breathe	2.7 μA/cm^2^	Ammonia sensing

**Table 3 sensors-24-00511-t003:** TENG in neuromuscular system real-time diagnosis.

Date	Sizes	Energy Sources	Performance	Applications
2021 [70]	3 cm × 3 cm	Vibration	29.8 mW/m^2^	Diagnosis of Parkinson’s disease
2022 [71]	5 cm × 5 cm	Vibration	26 mW	Diagnosis of Parkinson’s disease
2020 [72]	1 cm × 1 cm	Movement	141 μW	Finger motion sensing
2022 [73]	2 cm × 2 cm	Movement	20.2 V	Fall sensing
2022 [74]	2 cm × 3 cm	Vibration	1.33 W⋅m^2^	Motion diagnosis
2020 [75]	1 cm × 8 cm	Movement	0.1614 V/m	Fall sensing
2021 [76]	None	Movement	98 V	Bone morphology diagnosis
2022 [77]	None	Movement	9.2 V and 4.9 nA	Bone morphology diagnosis

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
