# Peer review of "From Body Monitoring to Biomolecular Sensing: Current Progress and Future Perspectives of Triboelectric Nanogenerators in Point-of-Care Diagnostics"

_sensors, 2024, doi:10.3390/s24020511_

Round 1

Reviewer 1 Report

Comments and Suggestions for Authors

The article is well written, however there are some typo errors which need to be corrected.

The comments are marked in the submitted article.

Author Response

Dear Reviewer,

Thank you very much for your valuable feedback and suggestions. I have carefully reviewed the comments in the PDF file you provided and have made amendments accordingly. For your convenience, these changes have been highlighted in the Word document attached (please ensure 'Track Changes' mode is enabled for a clear view of the modifications).

I would also like to draw your attention to significant revisions made in Chapter Five, which were based on the suggestions from other reviewers. Your further insights on these changes would be greatly appreciated.

Thank you once again for your thoughtful guidance and support in enhancing the quality of my work.

Sincerely, Zequan Zhao

Reviewer 2 Report

Comments and Suggestions for Authors

Overall, your review provides a comprehensive overview of the significance of triboelectric nanogenerators (TENGs) in the field of medical diagnostics. Here are a few minor comments to consider

What is the significance of TENGs in reducing the dependence of biosensors on external power sources in medical applications?

What is the basic principle behind how TENGs transform subtle physical movements into electrical energy, and how does this contribute to the development of self-powered biosensors?

What role do TENG-based biosensor designs play in achieving exceptional sensitivity and specificity in biomarker detection for accurate and efficient diagnoses?

Author Response

Thank you for providing valuable comments and suggestions regarding our manuscript. We appreciate the opportunity to enhance our work and address the points you raised. Below, we have outlined our responses to each comment and described the revisions made to the manuscript.

Comment 1: Significance of TENGs in Reducing Dependence on External Power Sources in Medical Applications

Response: We have elaborated on the significance of Triboelectric Nanogenerators (TENGs) in reducing the reliance of biosensors on external power sources within medical applications. We emphasized how TENGs facilitate sustainable energy harvesting from ambient mechanical movements, thus enhancing the portability and utility of biosensors, particularly in remote or resource-constrained environments.

Comment 2: Basic Principle of TENGs and Contribution to Self-Powered Biosensors

Response: The manuscript now includes a detailed explanation of the working principle of TENGs - the conversion of mechanical energy into electrical energy via triboelectrification and electrostatic induction. We have clarified how this principle is instrumental in developing self-powered biosensors that can autonomously operate and continuously generate energy for long-term monitoring.

Comment 3: Role of TENG-Based Biosensor Designs in Biomarker Detection

Response: We have expanded on how TENG-based biosensor designs contribute to achieving high sensitivity and specificity in biomarker detection. The revised manuscript details how these designs enable the detection of low-abundance biomarkers and can be tailored for specific biomarkers, which is crucial for early diagnosis and precise disease monitoring.

Manuscript Revisions

To further reflect the resolutions of these issues, we have optimized the overall structure of our manuscript. Specifically, we revised Chapter 5 to provide a clearer and more comprehensive discussion. Additionally, we incorporated four new tables that concisely present relevant data and elucidate the principles and applications of TENGs in biosensor technology. These modifications are intended to improve the manuscript's clarity, flow, and informational value, ensuring a thorough understanding of TENG-based biosensors in medical diagnostics.

We believe these revisions adequately address the comments and greatly enhance the manuscript's contribution to the field. We are grateful for your insightful feedback and hope that our revisions meet your expectations.

Thank you for your consideration, and we look forward to your further guidance.

Reviewer 3 Report

Comments and Suggestions for Authors

In this manuscript the authors aim to review TENG based point of care diagnostics. They discuss works related to TENG based human body monitoring and then works related to TENG based biochemical monitoring.

Here are the reviewers comments.

1. Some of the works mentioned are piezoelectric energy harvesters and not TENG such as Figure 7c, 7d, 10a, 10b. Only TENG related papers should be included as per the title of the manuscript.

2. The reviewer does not understand the point of Section 2. I suggest removing this section entirely and including some of the papers mentioned here, such as Figure 4b,c, in a later section on digestion and bladder.

3. If each section is restricted to just one figure it will make the paper better organized and readable. So all cardiac and respiratory papers could be just one figure instead of Figure 5,6,7. All papers related to neuromuscular sensing could be limited to just one figure instead of Figure 8,9. All papers related to biomolecular sensing could be limited to just one figure instead of Figure 10,11. Only the most comprehensive and impactful figures should be included, not Figures like 5d, 8c and 8d.

4. Please provide a nice summary table for the different PoCD categories, the specific issues that the papers address, and the TENG outputs associated.

Comments on the Quality of English Language

the English language is too flowery and extravagant at times instead of clear and to the point.

Author Response

Dear Reviewer,

Thank you for your valuable feedback on our manuscript. I have compiled my responses into a Word document for your review.

Reviewer 4 Report

Comments and Suggestions for Authors

This manuscript provides a comprehensive and insightful overview of the application of triboelectric nanogenerators (TENGs) in point-of-care diagnostics. The authors have effectively covered a wide range of topics, from the basic principles of TENGs to their application in protein-drug interactions and theranostic applications. The inclusion of recent advancements in the field and the use of conducting polymers like polyaniline (PANI) to enhance sensor performance demonstrates the paper's relevance to current research in the field of sensors and nanotechnology.

1. The paper would benefit from a more detailed discussion of the methodology used in the studies referenced. Providing insights into the experimental setups, data collection methods, and analysis techniques would enhance the paper's comprehensiveness.

2. It would be valuable to include a section discussing the limitations and challenges associated with the use of TENGs in point-of-care diagnostics. Addressing potential obstacles and areas for improvement would add depth to the review.

3. The paper could benefit from a more explicit discussion of the practical implications of the reviewed studies. How might the findings impact the development of real-world diagnostic devices and their integration into clinical settings?

Comments on the Quality of English Language

The use of language is precise, and the paper maintains a formal and professional tone throughout. The authors have also employed appropriate scientific vocabulary and terminology, contributing to the clarity and accuracy of the content.

Author Response

(The authors gave the same response as above.)

Round 2

Reviewer 3 Report

Comments and Suggestions for Authors

The authors have sufficiently addressed my concerns. The new tables and figures make the paper more organized and readable. The authors have also  removed the papers related to piezoelectric harvesters as requested and shortened Section 2.